# PeerJ

# With or without light: comparing the reaction mechanism of dark-operative protochlorophyllide oxidoreductase with the energetic requirements of the light-dependent protochlorophyllide oxidoreductase

Pedro J. Silva

REQUIMTE, Faculdade de Ciências da Saúde, Universidade Fernando Pessoa, Rua Carlos da Maia, Porto, Portugal

Corresponding author
Pedro J. Silva, pedros@ufp.edu.pt

## ABSTRACT

The addition of two electrons and two protons to the $C_{17}=C_{18}$ bond in protochlorophyllide is catalyzed by a light-dependent enzyme relying on NADPH as electron donor, and by a light-independent enzyme bearing a $(Cys)_3Asp$-ligated [4Fe–4S] cluster which is reduced by cytoplasmic electron donors in an ATP-dependent manner and then functions as electron donor to protochlorophyllide. The precise sequence of events occurring at the $C_{17}=C_{18}$ bond has not, however, been determined experimentally in the dark-operating enzyme. In this paper, we present the computational investigation of the reaction mechanism of this enzyme at the B3LYP/6-311+G(d,p)//B3LYP/6-31G(d) level of theory. The reaction mechanism begins with single-electron reduction of the substrate by the $(Cys)_3Asp$-ligated [4Fe–4S], yielding a negatively-charged intermediate. Depending on the rate of Fe–S cluster re-reduction, the reaction either proceeds through double protonation of the single-electron-reduced substrate, or by alternating proton/electron transfer. The computed reaction barriers suggest that Fe–S cluster re-reduction should be the rate-limiting stage of the process. Poisson–Boltzmann computations on the full enzyme–substrate complex, followed by Monte Carlo simulations of redox and protonation titrations revealed a hitherto unsuspected pH-dependence of the reaction potential of the Fe–S cluster. Furthermore, the computed distributions of protonation states of the His, Asp and Glu residues were used in conjunction with single-point ONIOM computations to obtain, for the first time, the influence of all protonation states of an enzyme on the reaction it catalyzes. Despite exaggerating the ease of reduction of the substrate, these computations confirmed the broad features of the reaction mechanism obtained with the medium-sized models, and afforded valuable insights on the influence of the titratable amino acids on each reaction step. Additional comparisons of the energetic features of the reaction intermediates with those of common biochemical redox intermediates suggest a surprisingly simple explanation for the mechanistic differences between the dark-catalyzed and light-dependent enzyme reaction mechanisms.

## INTRODUCTION

All life on Earth depends on the availability of reduced forms of carbon. As the reduction of simple carbon-containing molecules like $CO_2$ is a strongly endergonic process, additional sources of energy are needed to overcome this high thermodynamic hurdle. Although several organisms (collectively known as lithoautotrophs) are able to obtain that energy from the conversion of inorganic substances, the overwhelming majority of carbon reduction is performed by photosynthetic organisms, which obtain the necessary energy by capturing photons from visible light. These photons are used to excite chromophores, which then become highly efficient reducing species, ultimately providing both the low-potential electrons needed to reduce carbon and the ATP used by cells as energy-transfer molecule. The most abundant photosynthetic pigments, chlorophylls, are obtained from the tetrapyrrole protoporphyrin IX through a series of reactions that includes $Mg^{2+}$ complexation, methylation by an S-adenosylmethionine-dependent methyltransferase and six-electron oxidation, yielding an highly-unsaturated molecule, protochlorophyllide (PChlide), which absorbs mainly in the low-energy region of the spectrum and is therefore unable to drive the necessary charge separation in the photosynthetic reaction centers (*Masuda & Fujita, 2008*). Two different enzymes are able to increase the saturation of the PChlide ring and generate chlorophyllide (Chlide), a pigment that absorbs light in higher-energy regions of the spectrum: angiosperms contain an oxygen-insensitive light-dependent protochlorophyllide oxidoreductase (*Masuda & Takamiya, 2004*), whereas gymnosperms, algae and cyanobacteria possess an oxygen sensitive, dark-operating, protochlorophyllide oxidoreductase (*Fujita & Bauer, 2003*) evolutionarily related to nitrogenase.

The reaction mechanism of the light-dependent protochlorophyllide oxidoreductase has been extensively studied through experimental (*Heyes & Hunter, 2004*; *Heyes et al., 2009*; *Heyes et al., 2011*; *Sytina et al., 2012*) and computational (*Heyes et al., 2009*; *Silva & Ramos, 2011*) methods. In contrast, relatively little is known about the precise sequence of events taking place in the dark-operative protochlorophyllide oxidoreductase (dPCHOR). The enzyme contains two components: a homodimeric L-protein which performs ATP-dependent electron transfer reminiscent of that observed in nitrogenase Fe protein (*Fujita & Bauer, 2000*; *Sarma et al., 2008*), and a heterotetrameric component bearing the active site and a $(Cys)_3Asp$-ligated [4Fe–4S] cluster which accepts electrons from the L-protein and functions as the electron donor to the protochlorophyllide substrate (*Muraki et al., 2010*; *Bröcker et al., 2010*). The peculiar ligation of the electron-transferring [4Fe–4S] cluster has been shown by site-directed mutagenesis to be crucial to the enzyme activity (*Muraki et al., 2010*), probably due to the lowering of its reduction potential below that of other [4Fe–4S] clusters (*Kondo et al., 2011*; *Takano et al., 2011*). The crystallographic structure of dPCHOR (*Muraki et al., 2010*; *Bröcker et al., 2010*) shows that the substrate

**Figure 1 Overall reaction mechanism.** Comparison of the overall reaction mechanisms of light-dependent (central pathway) and dark-operative (bottom pathway) protochlorophyllide oxidoreductases. The rings have been labeled according to the IUPAC nomenclature (*Moss, 1987*).

binding site, while mostly lined by hydrophobic residues, contains a single conserved aspartate residue (Asp274) which is thought to be a proton-donor for the reaction. Two protons and two electrons are required (Fig. 1), which necessarily entails two separate reduction events (as the [4Fe–4S] cluster is a one-electron donor) and the presence of a second proton-donor. Asp274 is unlikely to act as the donor of the second proton, as it cannot be reprotonated due to the absence of pathways linking it to the solvent. The propionic acid side-chain present on the substrate $C_{17}$ was therefore proposed as the second proton donor (*Muraki et al., 2010*). The intricacies of proton and electron transfer from dPChOR to its protochlorophyllide substrate have, however, remained unaddressed by experimental methods. In this report, we describe this reaction mechanism with the help of density-functional theory methods. The results allow the description of the sequence of the reduction/protonation events and also identify the factors governing the stereochemical outcome of this enzyme-catalyzed reaction. Comparisons of the energetic features of the intermediates with those of common biochemical redox intermediates suggest a simple explanation for the differences observed in the dark-catalyzed and light-dependent enzyme reactions.

## COMPUTATIONAL METHODS

Coordinates of the active site were taken from the X-ray structure (3AEK) determined by *Muraki et al. (2010)*. Since the substrate binding cavity is almost completely lined with hydrophobic residues (which are generally inert from a reactional point of view) the computational model of the active site could be thoroughly pruned, to achieve a cost-effective computational model: computations with a very simplified substrate mimic (4-methyl-2,5-dimethylidene-2,5-dihydro-1H-pyrrol-3-yl)acetic acid, the proton-donating Asp274, and the second-shell amino acids Arg48 (which is located close to the proton-donating Asp274) and Gly409-Leu410 backbone (which may establish a single hydrogen bond to the propionic acid present on the substrate) showed that neglect of the second-shell amino acids affects proton-transfer energies by less than 2 kcal mol$^{-1}$. The

reaction mechanism was therefore studied using the complete natural substrate, the water molecule bound to its Mg atom, and the only amino acid side chain (Asp 274) able to interact with the substrate through hydrogen-bonds and/or proton donation. To prevent unrealistic movements of the simplified computational model, the protochlorophyllide Mg atom and the Asp274 $C_\alpha$ and $C_\beta$ carbon atoms were constrained to their crystallographic positions. Geometry optimizations were performed with the Firefly (*Granovsky, 2013*) quantum chemistry package, which is partially based on the GAMESS (US) (*Schmidt et al., 1993*) source code, at the B3LYP (*Lee, Yang & Parr, 1988*; *Becke, 1993*; *Hertwig & Koch, 1995*) level with the 6-31G(d) basis set, using autogenerated delocalized coordinates (*Baker, Kessi & Delley, 1996*). Transition states were located by scanning the appropriate reaction coordinates, taking the highest-energy point in these scans, and following the highest imaginary frequency computed at those geometries to the appropriate first-order saddle point. All transition states found contained an imaginary frequency connecting the reactant state to the product state of that reaction step and a few vibration modes with small imaginary frequencies due to the constrained atoms. Zero-point and thermal effects on the enthalpies/free energies at 298 K were computed at the optimized geometries using a scaling factor of 0.9804 (*Foresman & Frisch, 1996*). Single-point energies were computed with the triple-zeta 6-311G(d,p) basis set, augmented with a set of diffuse functions on the oxygen, henceforth called 6-311G(+)(d,p).

The Asp(Cys)$_3$-ligated Fe–S cluster was optimized separately in the reduced (charge $= -3$, spin $= 1/2$) and oxidized (charge $= -2$, spin $= 0$) forms, using the SBKJC effective core potential and associate basis set for Fe and 6-31G(d) for the other elements. The $C_\alpha$ and $C_\beta$ carbon atoms of the coordinating amino acids were constrained to their crystallographic positions to prevent unrealistic movements and to capture the subtle influence of the conformation of the cysteinyl side chains on the redox potential of the Fe–S cluster (*Niu & Ichiye, 2009*). Appropriate broken-symmetry initial guesses of the Fe–S cluster density were generated using a combination of the protocols of *Szilagyi & Winslow (2006)* and *Greco et al. (2010)*. Single-point energies of the optimized geometries of the Fe–S cluster were computed using the all-electron s6-31G* basis set (*Swart et al., 2010*) for Fe and 6-311G(2d,p) for all other elements. Intra- and inter-molecular dispersion effects were computed with the DFT-D3 formalism developed by *Grimme et al. (2010)*. The activation energy of the one-electron transfer between the reduced Fe–S cluster and the substrate was estimated by applying Marcus theory for electron transfer, as suggested by *Blomberg & Siegbahn (2003)*. Reorganization energies for the Fe–S cluster and substrate in each oxidation state were computed using the reactant geometry for the product state (e.g., the oxidized state energy is computed at the reduced Fe–S cluster geometry, etc.) and vice-versa. Activation energies were then computed by building appropriate Marcus parabolas using these reorganization energies, as shown in Fig. 2.

All energy values described in the text include solvation effects ($\varepsilon = 10$) computed using the Polarizable Continuum Model *Tomasi & Persico (1994)*, *Mennucci & Tomasi (1997)* and *Cossi et al. (1998)* implemented in Firefly. $\varepsilon = 10$ was chosen instead of the more common $\varepsilon = 4$ to model some of the stabilization of the ionic forms of Asp274 and propionic acid

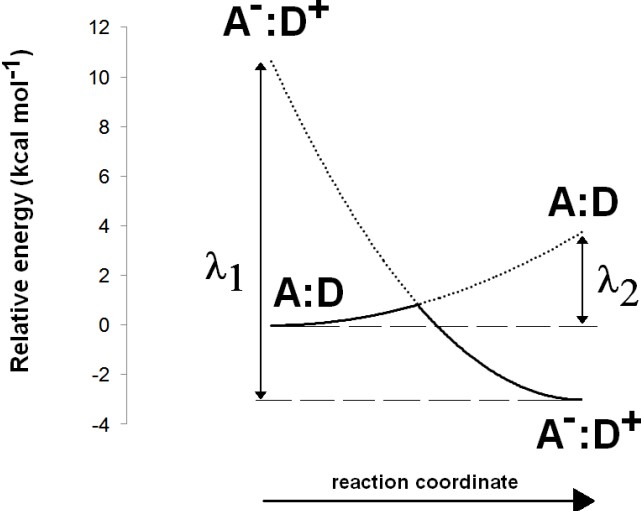

**Figure 2 Computation of electron transfer activation energies.** Determination of activation energies of electron transfer from a donor (D) to an acceptor (A). $\lambda_1$ is the reorganization energy of the $A^-:D^+$ complex, defined as the energy needed to bring the charge-separated complex to the geometry of the neutral complex A:D. $\lambda_2$ is the reorganization energy of the neutral A:D complex, defined as the energy needed to bring this complex to the geometry of the charge-separated complex $A^-:D^+$.

residues provided in the enzyme by hydrogen bonding with the Gly409-Leu410 backbone amide. Energies computed at other dielectric constants are shown in Table 1. In Table 6, energies of reactions involving addition of $n$ non-modeled solvent protons were computed as:

$$\Delta G = \Delta G_{\text{solvated products}} - \Delta G_{\text{solvated reactants}} - n\Delta G_{\text{solvated H+}}.$$

For the solvation free energy of $H^+$, $\Delta G_{\text{solv,H+}}$, we used the value of $-265.9$ kcal mol$^{-1}$, obtained by converting the experimental value of $-263.98$ kcal mol$^{-1}$ (*Tissandier et al., 1998*) to the appropriate thermodynamic standard state conventions as recommended by *Kelly, Cramer & Truhlar (2006)*. To enable direct comparison of energies with a different number of non-modeled solvent protons, these $\Delta G$ values were then converted to effective $\Delta G$ at pH = 7.0:

$$\Delta G_{\text{eff}} = \Delta G - RT\ln[H^+]^n.$$

Energy differences between $n$-electron containing species and the corresponding $n + 1$-electron-containing analogues were converted to reduction potentials ($\Delta E$) through

$$\Delta G = -nF\Delta E,$$

where $n$ is the number of electrons added to the species and $F$ is the Faraday constant (96,485 C mol$^{-1}$). In order to characterize the protonation states of Asp274 and the propionic acid side chain in the enzyme–substrate complex, continuum electrostatic calculations were performed using MEAD (*Bashford & Gerwert, 1992*). AMBER03 charges and radii (*Duan et al., 2003*) were assigned to the protein structure using YASARA

**Table 1 Relative enthalpies (in kcal mol$^{-1}$) of all intermediates and transition states in the reaction mechanism of light-independent protochlorophyllide oxidoreductase.** Energies were computed at the B3LYP-D3/6-311+G(d,p)//B3LYP/6-31G(d) level of theory.

| Added electrons | Proton on... | Proton on... | $\varepsilon = 4$ | $\varepsilon = 10$ | $\varepsilon = 20$ | $\varepsilon = 78.36$ |
|---|---|---|---|---|---|---|
| 0 | Asp274 | Propionate | 0.0 | 0.0 | 0.0 | 0.0 |
| 0 | Asp274 | Propionate●●●C18 | 24.0 | 24.7 | 25.0 | 25.2 |
| 0 | Asp274 | C18 | 19.7 | 19.8 | 19.9 | 19.9 |
| 0 | Asp274●●●C17 | Propionate | 31.3 | 31.4 | 31.5 | 31.6 |
| 0 | C17 | Propionate | 26.3 | 25.3 | 25.0 | 24.7 |
| 1 | Asp274 | Propionate | −62.2 | −67.4 | −69.3 | −70.7 |
| 1 | Asp274●●●C17 | Propionate | −49.1 | −53.8 | −55.5 | −56.8 |
| 1 | C17 | Propionate | −64.5 | −70.5 | −72.6 | −74.2 |
| 1 | Asp274 | Propionate●●●C18 | −53.3 | −58.3 | −60.0 | −61.3 |
| 1 | Asp274 | C18 | −68.5 | −73.9 | −75.9 | −77.3 |
| 1 | Asp274●●●C17 | C18 | −50.2 | −55.1 | −56.7 | −58.0 |
| 1 | C17 | Propionate●●●C18 | −54.2 | −59.7 | −61.6 | −63.0 |
| 1 | C17 | C18 | −68.4 | −75.6 | −78.1 | −79.9 |
| 1 | C18* | Propionate | −53.9 | −59.6 | −61.6 | −63.1 |
| 1 | Asp274 | C17* | −51.9 | −58.4 | −60.7 | −62.5 |
| 2 | Asp274 | Propionate | −101.5 | −121.2 | −128.1 | −133.4 |
| 2 | Asp274●●●C17 | Propionate | −94.6 | −112.7 | −119.0 | −123.8 |
| 2 | C17 | Propionate | −123.3 | −142.1 | −148.6 | −153.5 |
| 2 | Asp274 | Propionate●●●C18 | −98.2 | −116.4 | −122.8 | −127.6 |
| 2 | Asp274 | C18 | −119.2 | −138.7 | −145.5 | −150.7 |
| 2 | C17 | Propionate●●●C18 | −126.8 | −145.3 | −151.7 | −156.6 |
| 2 | Asp274●●●C17 | C18 | −124.9 | −143.7 | −150.2 | −155.1 |
| 2 | C17 | C18 | −160.1 | −179.1 | −185.6 | −190.5 |
| 2 | C18* | Propionate | −112.5 | −130.5 | −136.7 | −141.4 |
| 2 | Asp274 | C17* | −113.4 | −132.9 | −139.6 | −144.8 |

**Notes.**
ZPVE, dispersion and solvation effects at several dielectric constants ($\varepsilon$) are included. (*) Wrong stereochemistry on the protochlorophyllide $C_{17}$ or $C_18$ atoms. Proton-transfer transition states are labeled in the format X●●●Y (where X and Y are the atoms/residues donating and accepting the proton).

(*Krieger et al., 2004*). Substrate and Fe–S cluster charges were assigned according to the RESP protocol (*Bayly et al., 1993*). The solvent probe radius was 1.4 Å, which should provide a reasonable spherical approximation of the water molecule. The ionic exclusion layer thickness was set to 2.0 Å, and temperature at 300 K. The dielectric constant used for the solvent region was 80, the approximate value for bulk water at room temperatures. The dielectric constant for the protein interior was set to 15, the value previously found to yield optimum results with this methodology (*Antosiewicz, McCammon & Gilson, 1994*; *Martel et al., 1999*). A two-step focusing method was used. A first calculation using a (200 Å)$^3$ cube with a 1.0-Å lattice spacing, centered on the protein was followed by a second calculation using a (25 Å)$^3$ cube with a 0.25-Å spacing, centered on the titrable site. All Asp, Glu and His residues, as well as the substrate propionic acid substituent, were allowed to titrate. The sampling of proton-binding states was done using the MCRP program (Monte Carlo for Reduction and Protonation), which implements a Monte Carlo method

described by *Baptista, Martel & Soares (1999)* and *Teixeira, Soares & Baptista (2002)*. Initial sampling was performed at 0.1 pH units intervals in the 5–9 range and at 20 mV intervals from −750 mV to −200 mV using $2 \times 10^5$ Monte Carlo steps. In production runs, amino acids found to remain protonated or deprotonated over 90% (or more) of the sampling grid were kept at their protonated (or deprotonated) states, and the remaining sites were allowed to titrate freely for $1 \times 10^6$ Monte Carlo steps an each pH/electric potential point.

The electrostatic influence of the full protein on the active system ($\Delta_{\text{prot}}$) was analyzed using an ONIOM-inspired (*Dapprich et al., 1999*) methodology:

$$\Delta_{\text{prot}} = E_{\text{MM total system}} - E_{\text{MM active site}},$$

where $E_{\text{MM total system}}$ is total electrostatic energy of the protein + intermediate system) and $E_{\text{MM active site}}$ is the active site (Asp274 + (Cys)$_3$ Asp ligated 4Fe–4S cluster + intermediate) electrostatic energy. Each gas-phase-optimized intermediate was first superimposed on the crystal structure of the substrate in the substrate-bound enzyme. The conformation of its propionate/propionic acid substituent was then optimized through a brief steepest descent run to avoid any clashes with the rest of the protein, which was kept frozen. Charges on protein atoms were assigned according to the AMBER03 forcefield, whereas the charges on the intermediate, the Fe–S clusters and the Fe-coordinating residues were derived according to the RESP protocol. Currently, ONIOM computations (and other QM/MM approaches) always assume that the protonation states of the amino acids present in the portion of the molecule described by the molecular-mechanics force field remain fixed, with all amino acids with predicted $pK_a$ below the solution pH kept deprotonated and those with predicted $pK_a$ above the solution pH kept protonated. This approach is inevitable when studying the full reaction pathway in QM/MM framework due to the need to perform extensive sampling of the conformational space of the protein + substrate environment (*Klähn et al., 2005*; *Claeyssens et al., 2005*; *Kamerlin, Haranczyk & Warshel, 2009*; *Lonsdale, Harvey & Mulholland, 2010*; *Lonsdale et al., 2013*; *Rommel & Kästner, 2011*), but may introduce errors due to the possibility that other combinations of protonation states with similar population to this postulated state exist and afford more favorable electrostatic environments. In this work, the information gleaned from the Continuum Electrostatics computations described above was used to refine the ONIOM-derived energies by simultaneously considering all possible protonation states of the titrating amino acids. A phenomenological "average" electrostatic stabilization was computed as the Boltzmann-averaged electrostatic contribution of all possible protonation states of the protein:

$$e^{-\frac{\Delta E_{\text{average}}}{RT}} = \sum_{i=1}^{2^N} p_i e^{-\frac{\Delta E_i}{RT}},$$

where $N$ is the number of titrating acid/base amino acids, $p_i$ is the probability of a specific combination of protonation states in the population of $n$ acid/basic sites, $\Delta E_i$ is the electrostatic stabilization afforded by this combination of protonation states and the other symbols have their usual meanings. The energetic contributions of each (de)protonated

amino acid to the electrostatic stabilization energy are additive, and therefore $\Delta E_i$ can be computed directly for each combination of protonation states by simply summing the individual amino acid contributions In the absence of interactions between acid/base sites, $p_i$ could also be computed easily as the product of the individual probabilities of finding each site in the corresponding protonated/deprotonated state. Inter-site interaction prevents these probabilities from being computed directly, but a suitable estimate may be obtained by taking the distribution of protonation states sampled from the Monte Carlo simulations. The memory requirements of these simulations increase exponentially with the number of protonation sites which are tracked simultaneously, and therefore all sites that remained at least 90% (de)protonated from pH 5 to pH 9 and redox potential between $-750$ mV and $-200$ mV were taken as 100% (de)protonated to achieve computational tractability. The remaining forty-four sites were then divided into five/six groups to achieve manageable memory requirements. By grouping together the correlated amino acids, one can ensure that the correlations between the protonation states of the groups remain mostly negligible. The total "phenomenological" electrostatic stabilization can then simply be obtained from the addition of the partial phenomenological electrostatic stabilization energies of each of the six sub-groups:

$$
\begin{aligned}
e^{-\frac{\Delta E_{\text{average}}}{RT}} &= \sum_{i=1}^{2^N} p_i e^{-\frac{\Delta E_i}{RT}} \\
&= \sum_{i_1=1}^{2^{N_1}} \sum_{i_2=1}^{2^{N_2}} \sum_{i_3=1}^{2^{N_3}} \sum_{i_4=1}^{2^{N_4}} \sum_{i_5=1}^{2^{N_5}} \sum_{i_6=1}^{2^{N_6}} p_{i_1} p_{i_2} p_{i_3} p_{i_4} p_{i_5} p_{i_6} e^{-\frac{\Delta E_{i_1}+\Delta E_{i_2}+\Delta E_{i_3}+\Delta E_{i_4}+\Delta E_{i_5}+\Delta E_{i_6}}{RT}} \\
&= \sum_{i_1=1}^{2^{N_1}} \sum_{i_2=1}^{2^{N_2}} \sum_{i_3=1}^{2^{N_3}} \sum_{i_4=1}^{2^{N_4}} \sum_{i_5=1}^{2^{N_5}} p_{i_1} p_{i_2} p_{i_3} p_{i_4} p_{i_5} e^{-\frac{\Delta E_{i_1}+\Delta E_{i_2}+\Delta E_{i_3}+\Delta E_{i_4}+\Delta E_{i_5}}{RT}} \sum_{i_6=1}^{2^{N_6}} p_{i_6} e^{-\frac{\Delta E_{i_6}}{RT}} \\
&= \left( \sum_{i_1=1}^{2^{N_1}} \sum_{i_2=1}^{2^{N_2}} \sum_{i_3=1}^{2^{N_3}} \sum_{i_4=1}^{2^{N_4}} \sum_{i_5=1}^{2^{N_5}} p_{i_1} p_{i_2} p_{i_3} p_{i_4} p_{i_5} e^{-\frac{\Delta E_{i_1}+\Delta E_{i_2}+\Delta E_{i_3}+\Delta E_{i_4}+\Delta E_{i_5}}{RT}} \right) \times e^{-\frac{\Delta E_{\text{group 6}}}{RT}} \\
&= \cdots = e^{-\frac{\Delta E_{\text{group 1}}}{RT}} \times e^{-\frac{\Delta E_{\text{group 2}}}{RT}} \times e^{-\frac{\Delta E_{\text{group 3}}}{RT}} \times e^{-\frac{\Delta E_{\text{group 4}}}{RT}} \\
&\quad \times e^{-\frac{\Delta E_{\text{group 5}}}{RT}} \times e^{-\frac{\Delta E_{\text{group 6}}}{RT}} \\
\Leftrightarrow \\
\end{aligned}
$$

$$
\Delta E_{\text{average}} = \Delta E_{\text{group 1}} + \Delta E_{\text{group 2}} + \Delta E_{\text{group 3}} + \Delta E_{\text{group 4}} + \Delta E_{\text{group 5}} + \Delta E_{\text{group 6}}.
$$

The electrostatic stabilization due to the protein was computed for each proton- and electron-transfer step, and added (as a correction) to the electronic energy values obtained from gas-phase computations of the Fe–S cluster and substrate/intermediates at infinite distance (since gas phase DFT computations of the electronic state of the combined Fe–S cluster + substrate/intermediate system in the reduced-cluster state converged to unphysical solutions with an oxidized cluster and a super-reduced substrate). Similar computations in the presence of a PCM continuum show that using infinitely-separated subsystems introduces a small error ($<2$ kcal mol$^{-1}$) in the proton-transfer reaction steps

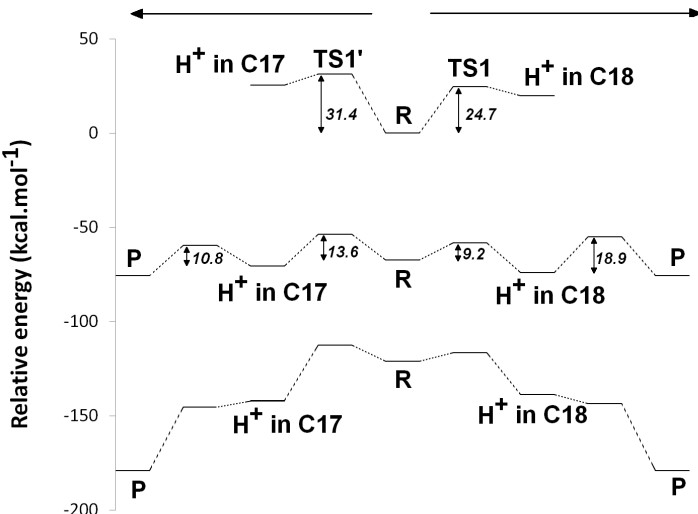

**Figure 3 Potential energy surfaces of proton and electron transfer events in light-independent PChOR, computed at the D3-B3LYP/6-311G(+)(d,p)//B3LYP/6-31G(d) level of theory with $\varepsilon = 10$.** The upper trace represents the potential energy surface (PES) with no added electrons; middle trace: PES at the one-electron-reduced state; lower trace: PES at the two-electron-reduced state. The energetic distance between the potential energy surfaces computed at different reduction states may be converted to redox potentials, as described in the methods section and discussed in the text.

and favors the electron-transfer steps by a larger amount (4–8 kcal mol$^{-1}$), particularly at lower values of $\varepsilon$.

## RESULTS AND DISCUSSION

In the crystal structure, Asp274 lies in the proper position to yield an intermediate with the correct stereochemistry only if it acts as proton donor to $C_{17}$. Therefore, we began our investigations assuming that Asp274 protonates $C_{17}$ and the propionic acid side chain protonates $C_{18}$. We also computed the reaction energetic for the proton transfers leading to the products with the wrong stereochemistries on $C_{17}$ and $C_{18}$.

### Proton-transfer events

The experimentally-obtained enzyme activity of dPChOR (*Muraki et al., 2010*) (50 nmol min$^{-1}$ mg$^{-1}$) sets an upper limit of 19.3 kcal mol$^{-1}$ for the rate-determining step of the overall process, using the well-known Eyring equation, $k_{cat} = \frac{k_B T}{h} e^{-\frac{\Delta G^{\ddagger}}{RT}}$, where $k_{cat}$ is the measured rate-constant, $k_B$ is the Boltzmann constant, $h$ is the Planck constant and $\Delta G^{\ddagger}$ is the activation free energy. The initial generation of a reaction intermediate from proton-transfer from Asp274 to $C_{17}$ can therefore be ruled out, as its energy lies 31.4 kcal mol$^{-1}$ above the reactant state (Fig. 3 and Table 1). In contrast, the computed barrier for the proton transfer from the propionic side chain to $C_{18}$ (24.7 kcal mol$^{-1}$) agrees reasonably well with the experimental value. The difference in computed stabilities between the $C_{17}$- and $C_{18}$-protonated isomers is more pronounced in the gas phase, which shows that the most important factor favoring the $C_{18}$- over the $C_{17}$- isomer is of

an electrostatic nature. Indeed, although both systems contain a positive charge in the substrate aromatic ring and a negative charge on a carboxylate group, the distance between these charges is much larger in the $C_{17}$-isomer/Asp274 carboxylate system.

In the one-electron-reduced state, the excess spin is, as expected, strongly delocalized across the porphyrin $\pi$-system ($\approx 0.1$ spin each on rings B and D, $\approx 0.25$ on ring A, $\approx 0.30$ on the C/E rings, and the remaining spin on the methylene bridges). In this state, the proton uptake becomes much more favorable than before (by $> 15$ kcal mol$^{-1}$, irrespective of the protonation site), due to the formation of a neutral protochlorophyllide/negative carboxylate intermediate, rather than the charge-separated pair observed in the non-reduced state. Proton-transfer to $C_{18}$ is predicted to occur with a very small barrier of 9.2 kcal mol$^{-1}$, whereas the proton-transfer from Asp274 to $C_{17}$ has a larger barrier of 13.6 kcal mol$^{-1}$. This difference in barriers seems to be correlated to the proton/protochlorophyllide distance observed in the transition state (1.46 Å for the Asp274-$C_{17}$ transfer vs. 1.31 Å for the carboxylate sidechain-$C_{18}$ transfer). The transfer of an additional proton to the one-electron-reduced/singly-protonated substrate may then occur with moderate barriers. The transfer from the propionic side chain to $C_{18}$ is again faster than that of Asp274 to C17 (10.8 kcal mol$^{-1}$, vs. 18.9 kcal mol$^{-1}$). The total barrier for the two consecutive proton-transfer events at the one-electron-reduced is fully consistent with the experimental value, regardless of the precise sequence of these events (13.6 kcal mol$^{-1}$ for Asp274-$C_{17}$ transfer followed by propionic sidechain-$C_{18}$ transfer; 18.9 kcal mol$^{-1}$ for the sequence initiated with transfer to $C_{18}$).

In the two-electron-reduced state, the barrier for the $Asp_{274}$-$C_{17}$ proton transfer (8.5 kcal mol$^{-1}$) is almost as low as the barrier for the proton transfer from the propionic sidechain to the $C_{18}$ atom (4.8 kcal mol$^{-1}$). Transfer of the second proton to the ring occurs without an enthalpic barrier in both cases, yielding the chlorophyllide product with the deprotonated Asp274 and propionate sidechain.

Analysis of alternative protonation events was also performed, to ascertain the reasons behind the observed stereochemical outcome. These computations showed that proton transfer from the propionic acid side chain to $C_{17}$ (yielding the wrong configuration in this carbon atom) is less favorable than any of the stereochemically correct proton transfers (Asp274 to $C_{17}$ and propionic acid to $C_{18}$), even in our simplified models which do not include the full steric constraints imposed by the hydrophobic amino acids lining the active site. The difference in total energies amounts to 11 kcal mol$^{-1}$ in the one-electron-reduced state, and to 8.5 kcal mol$^{-1}$ at the two-electron-reduced state. This intermediate contains (especially at the one-electron-reduced state) an unfavorable steric interaction, as the wrong configuration at $C_{17}$ pushes the propionate side chain towards Asp274 (Fig. 4). Protonation of $C_{18}$ by Asp274, which generates the wrong configuration on $C_{18}$, is also less favorable by 16 kcal mol$^{-1}$ at the one-electron-reduced state, and by 9.5 kcal mol$^{-1}$ at the two-electron-reduced-state. This destabilization occurs in spite of the lack of sterical clashes because no favorable stabilizing interactions of the carboxylate in Asp274 with the $Mg^{2+}$-coordinating water are possible in this isomer, in contrast to the correct isomer that arises when the carboxylate forms on the propionic acid side chain.

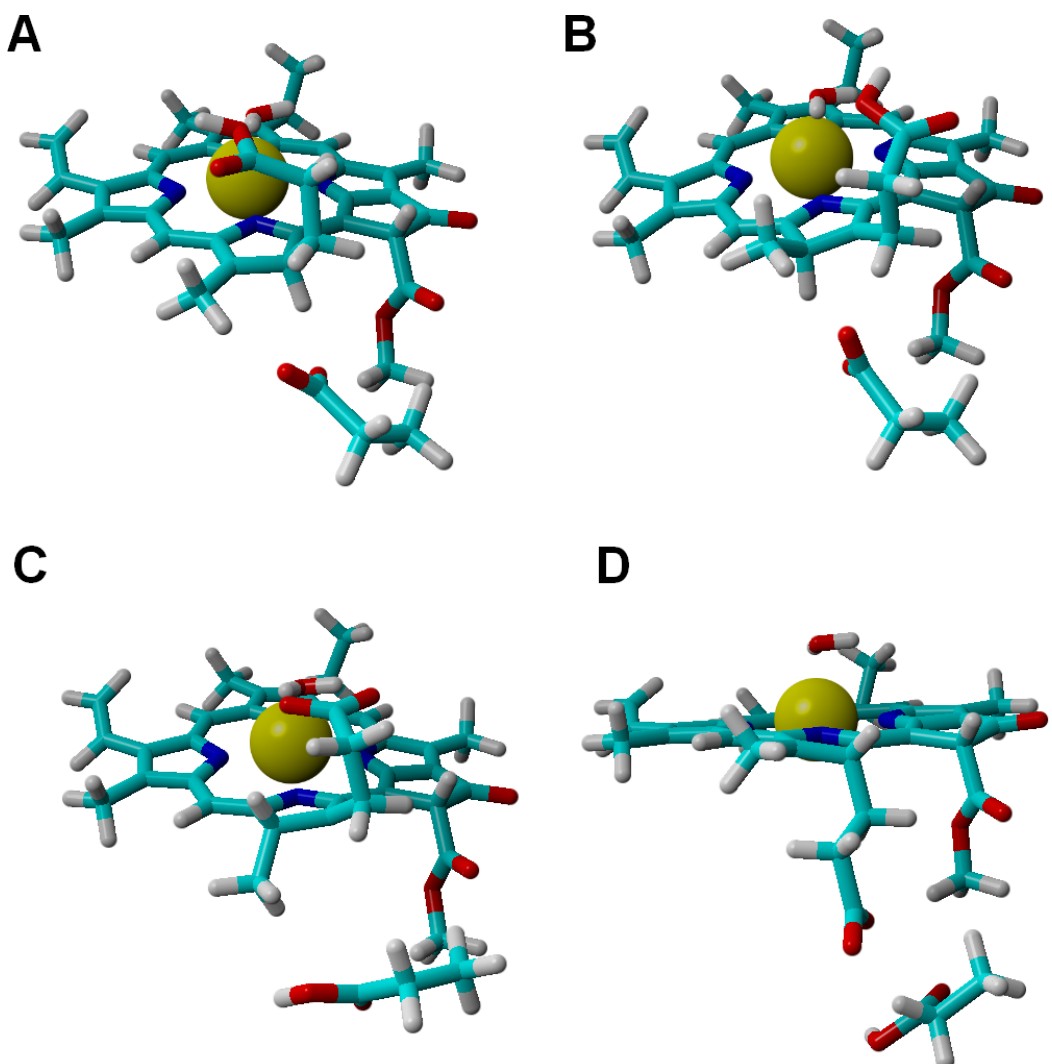

**Figure 4 Structures of intermediates with a singly-protonated $C_{17}=C_{18}$ bond at the one-electron-reduced state.** Comparison of the structures bearing a singly-protonated $C_{17}=C_{18}$ bond at the one-electron-reduced state. (A) Bond protonated on $C_{17}$ by Asp274 (correct stereochemistry). (B) Bond protonated on $C_{17}$ by the propionic acid side chain (wrong stereochemistry). (C) Bond protonated on $C_{18}$ by the propionic acid side chain (correct stereochemistry). (D) Bond protonated on $C_{18}$ by Asp274 (wrong stereochemistry).

## Electron-transfer events

The ease of reduction of each intermediate may be easily computed by taking the difference of energies between any $n$-electron containing species and the corresponding $n + 1$-electron-containing analogues. It can readily be seen (Table 2, last column), that one-electron reduction of the reaction intermediates depends strongly on the charge present on the PChlide ring: species with negative or neutral protochlorophyllide rings have an absolute redox potential between 2.3 and 3.2 V, whereas intermediates bearing one or more positive charges have much more favorable absolute redox potentials between 4.0 V and 4.5 V. In dark-operating PChOR, the electron-donating species is an unusual

**Table 2 Relative enthalpies (in kcal mol$^{-1}$) of intermediates in the reaction mechanism of light-independent PChOR in the presence of (independently optimized) [4Fe–4S] cluster.** Energies were computed at the B3LYP-D3/6-311(+)G(d,p)-s6-31G* (Fe)-6-311G(2d,p) (non-Fe atoms in the Fe–S cluster)//B3LYP/6-31G(d) level of theory. Solvation effects at $\varepsilon = 10$ are included.

| Electrons added to the substrate | H$^+$ in | H$^+$ in | Relative energy (with reduced cluster) | Relative energy (with oxidized cluster) | Fe–S redox potential (V) | Substrate redox potential (with reduced cluster) (V) | Substrate redox potential (with oxidized cluster) (V) | Substrate redox potential (in the absence of cluster) (V) |
|---|---|---|---|---|---|---|---|---|
| 0 | Asp274 | Propionate | 0.00 | 0.00 | 2.70 | 2.78 | 2.81 | 2.92 |
| 0 | Asp274 | C18 | 20.34 | 19.98 | 2.69 | 3.90 | 3.91 | 4.06 |
| 0 | C17 | Propionate | 25.36 | 25.25 | 2.70 | 3.95 | 3.97 | 4.15 |
| 1 | Asp274 | Propionate | −64.05 | −64.70 | 2.67 | n.d. | n.d. | 2.33 |
| 1 | Asp274 | C18 | −69.64 | −70.17 | 2.68 | 2.47 | 2.49 | 2.81 |
| 1 | C17 | Propionate | −65.73 | −66.24 | 2.68 | 2.83 | 2.85 | 3.11 |
| 1 | C17 | C18 | −72.20 | −72.14 | 2.70 | 4.22 | 4.25 | 4.49 |
| 2 | Asp274 | C18 | −126.58 | −127.57 | 2.66 | n.a. | n.a. | n.a. |
| 2 | C17 | Propionate | −131.10 | −132.04 | 2.66 | n.a. | n.a. | n.a. |
| 2 | C17 | C18 | −169.60 | −170.10 | 2.68 | n.a. | n.a. | n.a. |

**Notes.**

n.d., not determined; n.a., not applicable, since at this stage the substrate cannot accept another electron.

(Cys)$_3$Asp-ligated [4Fe–4S] cluster, which has been assigned a redox potential of 3.1 V in previous computations (*Takano et al., 2011*). Since those computations were performed without geometrical constrains and with very truncated cysteine models (SCH$_3$) which do not allow the evaluation of the influence of the side chain geometry on the electronic properties of the clusters (*Niu & Ichiye, 2009*), we performed additional optimizations of the Fe–S cluster using ethanethiol as model for the cysteine side chains and appropriate constraining of their carbon atoms to their crystallographic positions. The absolute redox potential of the electron-donating (Cys)$_3$Asp-ligated [4Fe–4S] cluster in PChOR is thus computed to be 2.80 V, which implies that, in the absence of significant interactions between cluster and substrate, all electron-uptake events by protochlorophyllide (except those by the 1-electron-reduced substrate or by the 1-electron-reduced, C$_{18}$-protonated substrate) should be thermodynamically favorable, as electrons spontaneously move from the species with lower redox potentials to the ones with higher potentials. Additional single-point computations were then performed in models including the separately-optimized Fe–S clusters and reaction intermediates at their crystallographic positions to ascertain the mutual influence of the Fe–S electronic distribution on the redox potential of the reaction intermediates, and vice-versa (Table 2). Irrespective of the redox state of the Fe–S cluster, the reaction intermediates become harder to reduce by 0.1–0.15 V in the original, non-reduced, state, and by 0.25–0.4 V in the one-electron-reduced state. The electric dipoles of the reaction intermediates in turn lower the Fe–S cluster redox potential (i.e., facilitate its oxidation) by similar modest amounts (0.1–0.15 V), irrespective of the redox state of the substrate. These data also show that the oxidation state of the Fe–S cluster barely affects the energetics of the substrate protonation reactions and, consequently, should also barely affect their activation barriers.

The electronic energies of the combined Fe–S cluster/active site systems show that electron-transfer from the reduced Fe–S cluster to the protochlorophyllide intermediates is thermodynamically favored in almost all cases (Table 2), except for the reduction of the one-electron-reduced/C$_{18}$-protonated species , which is unfavorable by a few kcal mol$^{-1}$. For this thermodynamically disfavored reduction step no activation energy could be computed through the Marcus formalism as the parabolas do not touch (Supplemental Information), but comparisons with the reorganization energies at lower dielectric constants (at which this transfer becomes spontaneous) suggest that the barrier should not be too different from the others. For the spontaneous steps the electronic reorganization energies of the combined Fe–S cluster/active site systems are quite low, yielding activation energies below 4 kcal mol$^{-1}$, which entails that) reduction will generally be much faster than the protonation events, which were shown above to have activation energies in excess of 10 kcal mol$^{-1}$. Electron transfer from the reduced Fe–S cluster should therefore precede each protonation event, though only if the rate of re-reduction of the Fe–S cluster by the ATP-dependent L-protein (*Kondo et al., 2011*) does not become limiting. Two possible pathways emerge from this analysis, both arising from an initial one-electron transfer to the protochlorophyllide. In one of them (Fig. 5A), re-reduction of the Fe–S cluster is slower than any of the proton-transfer events at the one-electron reduced

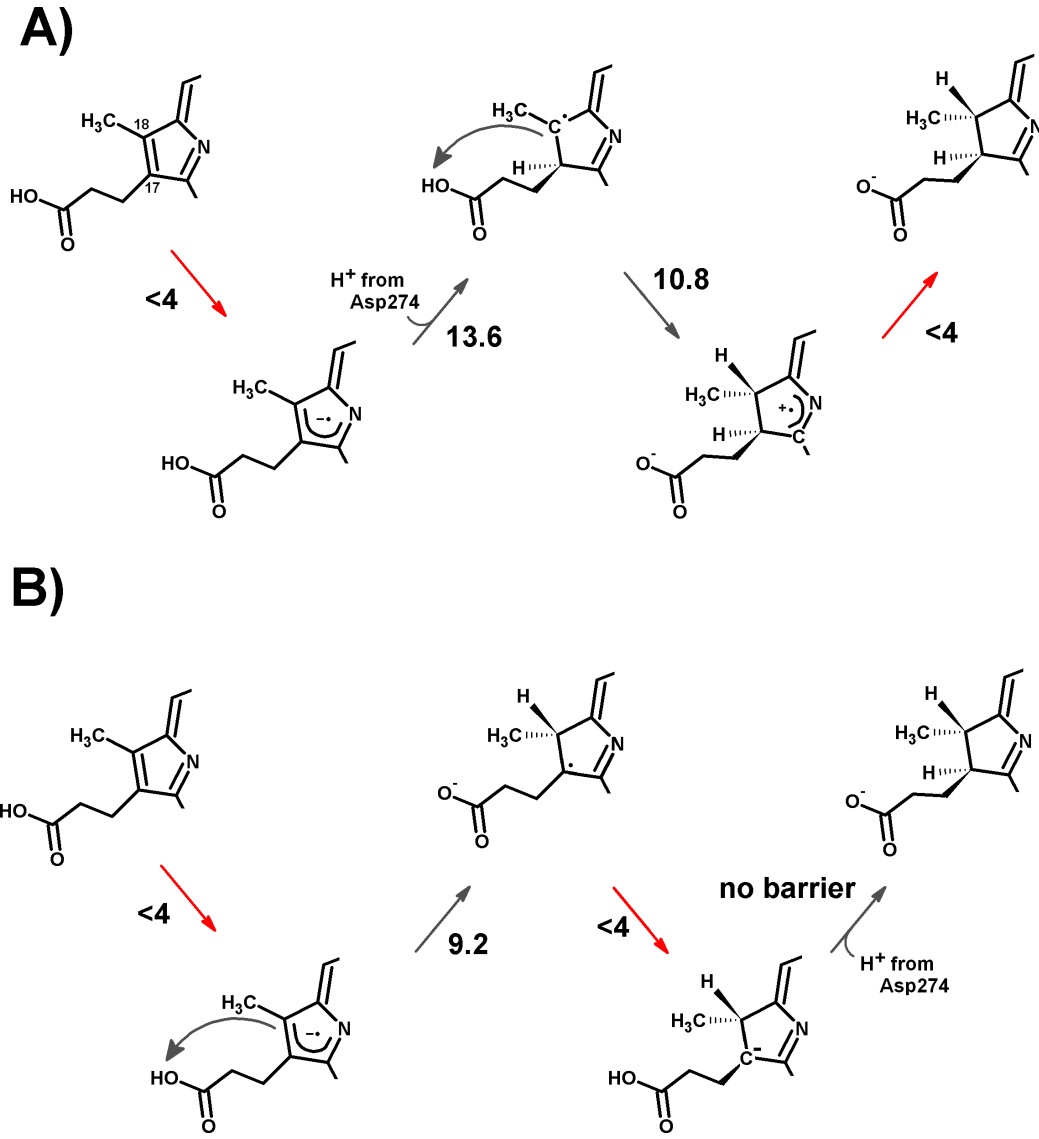

**Figure 5 Schematic representation of the most-favored computed reaction mechanisms.** (A) Slow Fe–S cluster re-reduction. (B) Fast re-reduction of the Fe–S cluster. Red arrows show the electron-transfer steps. Relevant activation energies (in kcal mol$^{-1}$) are shown for each reaction step. For simplicity, only the D ring of protochlorophyllide is represented.

state, which leads to the generation of a doubly protonated intermediated, preferably through the initial protonation at $C_{17}$, which has a lower overall barrier (13.6 kcal mol$^{-1}$) than the double-protonation starting with the $C_{18}$ atom (18.9 kcal mol$^{-1}$, as discussed earlier). After both protonations and cluster re-reduction occurs, electron transfer is both spontaneous and quite fast. In the second alternative (Fig. 5B) re-reduction of the Fe–S cluster is not rate-limiting and the second electron transfer to the substrate may occur immediately after the first protonation: in this instance, the reaction will likely proceed through protonation of $C_{18}$ by the propionic acid substituent of the protochlorophyllide D-ring, followed by electron transfer and barrier-less transfer of the second proton from

Asp274 to $C_{17}$. This pathway affords a barrier below 10 kcal mol$^{-1}$ and reaction rates far in excess of those observed experimentally, suggesting that Fe–S cluster re-reduction should indeed be the rate-limiting stage of the process.

## Estimating the influence of the protein environment on the reaction energetics

In the research mentioned in the previous sections, the inclusion of the full substrate binding-pocket was prevented by the unfavorable scaling of the computational cost of the high-levels of quantum theory used. Additionally, the extensive conjugation of the substrate $\pi$-system prevented the "sacrifice" of any portion of the substrate in the model for the sake of including more surrounding amino acids. Such exclusion should not affect the chemical steps since the omitted amino acids are relatively inert chemically due to their hydrophobic side chains and their influence is therefore only felt on the relative solvation of the reaction intermediates. The data computed at different dielectric constants (which is affected by the number and distribution of polar/apolar amino acids surrounding the substrate) show that the dependence of the reaction pathway with this factor is quite small. These observations agree with the large body of research (reviewed in *Himo & Siegbahn, 2003*; *Shaik et al., 2005*; *Ramos & Fernandes, 2008*; *Siegbahn & Blomberg, 2010*) which has established that the application of quantum chemical techniques to small-to-medium-size models of enzyme active sites can be extremely powerful in the thorough analysis of reaction pathways, provided that the charge distribution in the model accurately mimics that of the active site.

Several important characteristics of the reaction mechanism cannot be derived from truncated active site models due to the lack of the protein-induced electrostatic field, which depends on the overall charge distribution in the protein (*Stephens, Jollie & Warshel, 1996*; *Kamerlin & Warshel, 2010*; *Ribeiro, 2013*). In the enzyme studied in this report, such characteristics include the likelihood of occurrence of the active protonated states of Asp274 and propionic side chain at physiological pH and the susceptibility of the Fe–S cluster redox potential to the solution pH. Continuum electrostatics computations on the protochlorophyllide oxidoreductase structure bearing each of the quantum-chemically-optimized intermediate allowed us to quantify these effects (see Computational Methods for details). The inclusion of the protein barely affects the way the redox potentials of the Fe–S cluster vary as the intermediate gains electrons/protons (Table 3), but has an important effect on the sensitivity of the Fe–S cluster redox state towards changes in pH: in all instances the predicted change in redox potential per pH unit corresponds to the uptake of 0.8 protons by the Fe–S surroundings upon the one-electron reduction of the cluster. Analysis of the correlations matrixes clearly shows that the reduction of the Fe–S clusters increases the probability of finding the neighboring His53A and His13B in their protonated states. At lower pH, Asp147A also tends to become protonated as the cluster is reduced (Fig. 6).

The probability $p$ of finding the postulated proton-donating moieties (Asp274 and the propionic acid substituent) in their protonated states can also be computed using

**Table 3 Redox potentials (mV) of the Fe–S cluster proximal to the active site.** Potentials computed from the populations observed in Monte Carlo simulation of simultaneous redox/protonation events of the intermediate-bound light-independent protochlorophyllide oxidoreductase. Potentials are computed vs. an arbitrary internal reference, and therefore only relative changes of potentials (rather than absolute values) should be compared to experimental observations (*Martel et al., 1999*; *Teixeira, Soares & Baptista, 2002*).

| | | pH | | | | | | | | | | |
|---|---|---|---|---|---|---|---|---|---|---|---|---|
| Extra electrons in substrate | | 0 | 0 | 0 | 1 | 1 | 1 | 1 | 2 | 2 | 2 | 2 |
| $H^+$ on $C_{17}$ | | | Y | | | Y | | Y | | Y | | Y |
| $H^+$ on $C_{18}$ | | | | Y | | | Y | Y | | | Y | Y |
| No protein included | 5.0 | −430 | −423 | −425 | −440 | −432 | −433 | −425 | −449 | −441 | −442 | −433 |
| | 7.0 | −437 | −428 | −428 | −447 | −438 | −438 | −428 | −457 | −449 | −449 | −440 |
| | 9.0 | −437 | −429 | −428 | −448 | −439 | −438 | −428 | −459 | −449 | −450 | −439 |
| Protein included | 5 | −289 | −284 | −285 | −296 | −291 | −292 | −285 | −303 | −298 | −299 | −292 |
| | 7 | −398 | −390 | −394 | −405 | −399 | −401 | −393 | −414 | −408 | −410 | −402 |
| | 9 | −491 | −478 | −480 | −506 | −495 | −497 | −480 | −514 | −505 | −511 | −499 |
| | Slope (mV/pH unit) | −50.5 | −48.5 | −48.7 | −52.5 | −51 | −51.2 | −48.7 | −52.7 | −51.7 | −53 | −51.7 |

**Table 4 Energetic barrier increases (kcal mol$^{-1}$) at pH = 7.0 caused by the non-unitary probability of finding Aps274 and the propionic acid in the appropriate protonated states.**

| Reactant | Asp274 must be... | Propionic acid sidechain must be... | Probability of finding these protonation states | Energetic barrier increase (kcal mol$^{-1}$) |
|---|---|---|---|---|
| PChlide | Protonated | Protonated | $1.0 \times 10^{-5}$ | 6.9 |
| one-electron-reduced PChlide | Protonated | Protonated | $4.1 \times 10^{-4}$ | 4.7 |
| one-electron-reduced, $C_{17}$-protonated PChlide | Deprotonated | Protonated | $8.2 \times 10^{-4}$ | 4.2 |
| one-electron-reduced, $C_{18}$-protonated PChlide | Protonated | Deprotonated | $4.6 \times 10^{-3}$ | 3.2 |
| one-electron-reduced, $C_{17 \text{ and } 18}$-protonated PChlide | Deprotonated | Deprotonated | $9.95 \times 10^{-3}$ | 0.0 |
| two-electron-reduced PChlide | Protonated | Protonated | $7.7 \times 10^{-3}$ | 2.9 |
| two-electron-reduced, $C_{17}$-protonated PChlide | Deprotonated | Protonated | $1.1 \times 10^{-2}$ | 2.7 |
| two-electron-reduced, $C_{18}$-protonated PChlide | protonated | Deprotonated | $2.6 \times 10^{-2}$ | 2.2 |

these techniques, and converted into energetic barrier increases through the expression $\Delta\Delta G = -RT\ln p$ (Table 4). The predicted 6.9 kcal mol$^{-1}$ increase in the barrier of the initial step further prevents the reaction from proceeding through an initial protonation step, but does not prevent the one-electron reduction of the substrate due to its high energetic driving force and low (<4 kcal mol$^{-1}$) barrier. The barriers for subsequent steps increase by smaller values, so that both mechanisms depicted in Fig. 5 remain possible even after taking account the variable probabilities of finding Asp274 and propionic acid sidechain in the appropriate protonation states.

Additional effects of the assymetric protein-induced electrostatic field on the reaction energetics were analyzed using the ONIOM-inspired methodology described in the Methods section. The inclusion of all possible protonation states of the titrating amino

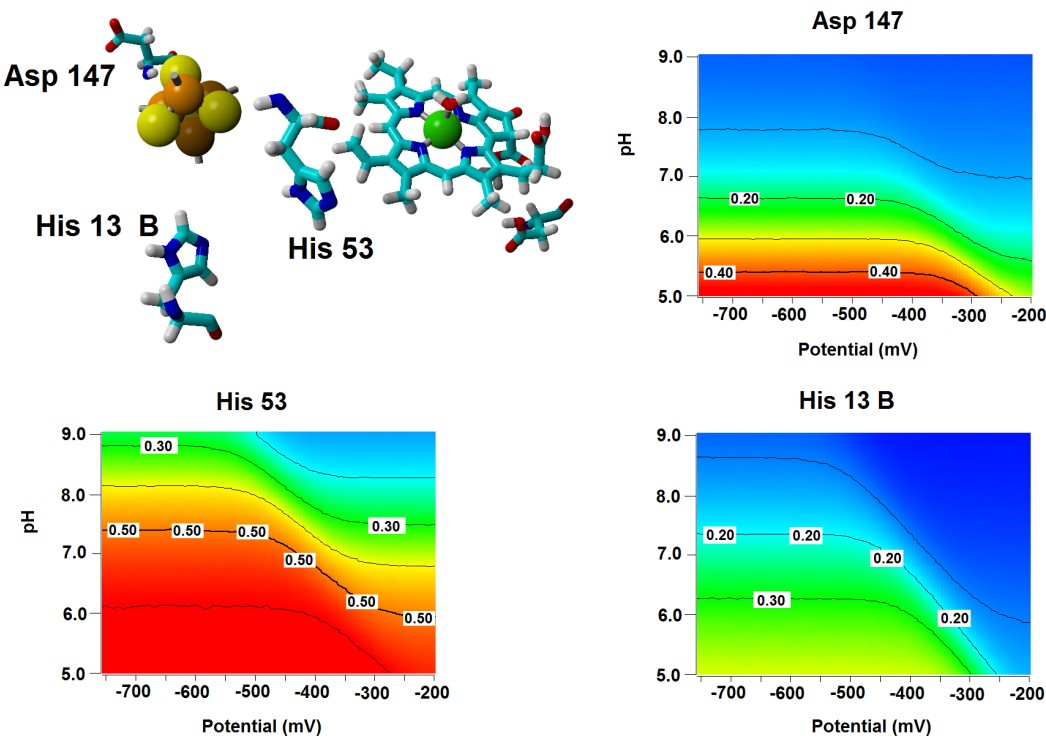

**Figure 6 Influence of the Fe–S cluster reduction state on the protonation of neighboring amino acids.** Arrangement of Asp147, His53 and His13B around the Fe–S cluster in the substrate-bound protein and protonation probabilities of these amino acids at different pH/electric potential.

acids on the computation of the "average" electrostatic field proved to be crucial, as several of the amino acids whose protonation state showed the higher electrostatic effects on the reaction energetics would otherwise be assumed to be present in their non-protonated states: for example, histidines 404B and 404D (present in the interface between subunits B and D, *ca.* 21 Å from the substrate) favor electron-transfer from the cluster to the substrate by 5–7 kcal mol$^{-1}$, but their effect would have been neglected by a computation that only took account of the most likely protonation state of each amino acid, as each remains (on average) 1/3 protonated in the pH/potential windows studied. Other amino acids whose protonation strongly favors electron-transfer are His 35, His 288, and His 378B. Protonation of His 86, His 13B, His 31B, His 64B and Asp 147, in turn, tend to disfavor this electron transfer. The presence of a protonated His378B favors the proton-transfer from the propionic acid side chain in the substrate to C$_{18}$, whereas protonation of His35 and His53 disfavors the proton-transfer from Asp274 to C$_{17}$ (Fig. 7 and Supplemental Information).

The energetic profiles of the proton-transfer-steps computed with the isotropic PCM model (Table 2) broadly agrees with the profile computed with the ONIOM-based correction to the gas-phase energies of infinitely-separated Fe–S cluster and substrate intermediates (Table 5), as proton-transfer from the propionic acid side chain to C$_{18}$ is consistently found to be thermodynamically more favorable than the transfer of Asp274

**Table 5 Relative energies (in kcal mol$^{-1}$) of intermediates in the reaction mechanism of PChOR in the presence of (independently optimized) [4Fe–4S] cluster, computed with an ONIOM-based methodology.**

| Electrons added to the substrate | H$^+$ in | H$^+$ in | Relative energy |
|---|---|---|---|
| 0 | Asp274 | Propionate | 0.00 |
| 0 | Asp274 | C18 | 23.5 |
| 0 | C17 | Propionate | 37.8 |
| 1 | Asp274 | Propionate | −175.2 |
| 1 | Asp274 | C18 | −182.6 |
| 1 | C17 | Propionate | −166.4 |
| 1 | C17 | C18 | −184.06 |
| 2 | Asp274 | Propionate | −288.9 |
| 2 | Asp274 | C18 | −310.3 |
| 2 | C17 | Propionate | −307.6 |
| 2 | C17 | C18 | −355.3 |

to C$_{17}$. Large differences are, however, observed in the electron-transfer steps, which are predicted by the ONIOM-based approach to be much more favorable (by *ca.* 115 kcal mol$^{-1}$ for the first electron moving to the substrate and by 25–30 kcal mol$^{-1}$ for the second electron), though their relative magnitudes closely follow the trends predicted by PCM. The exaggerated exergonicity afforded by the ONIOM-based computations are surely artifactual, as they would imply extremely high redox potentials for the substrate/intermediates: for example, a value of 3.15 V above the standard hydrogen electrode is predicted for PChlide, which is higher than the experimental values of the very strong oxidants fluorine (2.87 V) and ozone (2.07 V). This artifact most likely arises from the neglect of the relaxation of the protein upon protonation of its amino acids and of the surrounding water shell, which has been shown (*Schutz & Warshel, 2001*) to require the use of a higher "effective dielectric constant" to obtain accurate electrostatic stabilization energies. "True" electrostatic stabilization energies should be obtainable by dividing the value computed with $\varepsilon = 1$ (as in this work) by this "effective" dielectric constant, whose magnitude is site-dependent (*Schutz & Warshel, 2001*) and not immediately accessible from first-principles considerations. Interestingly, the Fe–S cluster (which lies buried inside the protein and far from the high-dielectric environment of the water solution) is predicted by the ONIOM-based methodology to have a much more reasonable redox potential (0.10 V vs. the standard hydrogen electrode) than PChlide, which lies much closer to the protein surface and whose redox potential should therefore be much more sensitive to the neglect of water in the computations. Indeed, screening the electrostatic stabilization of the Fe–S cluster with $\varepsilon = 1.06$ is enough to yield a computed redox potential of −0.32 V, in agreement with −0.4 V to −0.3 V range deduced from the redox potential of the Mg-ATP-activated Fe-cluster present in the nitrogenase Fe protein (*Ryle, Lanzilotta & Seefeldt, 1996*) which is known to be related to the L-protein that acts as electron-donor to PChOR.

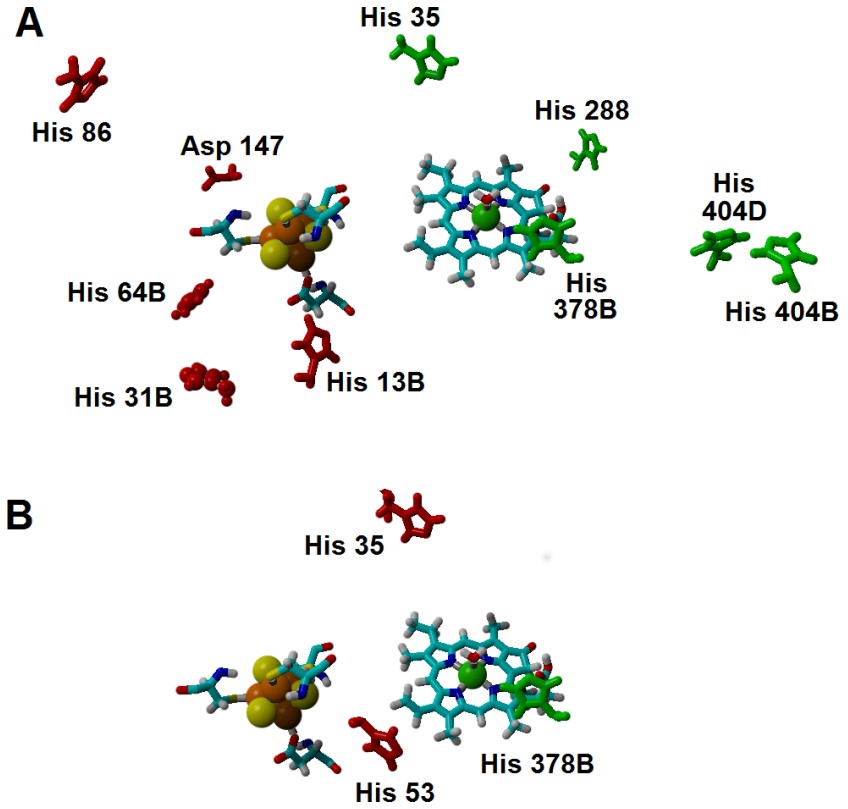

**Figure 7 Amino acids with strong influence on the proton-and electron-transfer reaction energies.** Amino acids which strongly affect the reaction energetics of (A) the electron-transfer steps or (B) the proton-transfer steps to $C_{17}$ and $C_{18}$. Favourable interactions are depicted in green, unfavourable interactions are shown in red. Amino acids which remain >85% protonated from pH 5 to pH 9 are depicted as ball-and-sticks.

**Table 6 Absolute redox potentials (V) of relevant redox pairs, computed at the DB3LYP-D3/6-311+G(d,p)//B3LYP/6-31G(d) level of theory.** Solvation effects and corrections for pH = 7.0 are included. "apoPChlide" and "apoChlide" refer to PChlide and Chlide devoid of $Mg^{2+}$. Despite ongoing controversy (*Donald et al., 2008*), the absolute reaction potential of the standard hydrogen electrode in water is usually taken as 4.43 V (*Reiss & Heller, 1985*).

| | Redox half-reaction | $\varepsilon = 4$ | $\varepsilon = 10$ | $\varepsilon = 20$ | $\varepsilon = 78.36$ |
|---|---|---|---|---|---|
| (a) | $NADPH^{2+} + 2\,e^- \rightarrow NADPH$ | 5.61 | 5.05 | 4.87 | 4.73 |
| (b) | $C_{18} -$ Protonated $PChlide + 2\,e^- + 1\,H^+ \rightarrow Chlide$ | 5.36 | 5.29 | 5.27 | 5.25 |
| (c) | Fumaric acid $+ 2\,e^- + 2\,H^+ \rightarrow$ succinic acid | 4.97 | 4.98 | 4.98 | 4.98 |
| (d) | $PChlide + 2\,e^- + 2\,H^+ \rightarrow Chlide$ | 4.93 | 4.94 | 4.94 | 4.94 |
| (e) | $apoPChlide + 2\,e^- + 2\,H^+ \rightarrow apoChlide$ | 4.53 | 4.53 | 4.53 | 4.53 |
| (f) | $NADP^+ + 2\,e^- + 1\,H^+ \rightarrow NADPH$ | 4.57 | 4.42 | 4.37 | 4.33 |
| (g) | $PChlide + 2\,e^- + 1\,H^+ \rightarrow$ deprotonated $Chlide$ | 3.60 | 3.72 | 3.76 | 3.79 |
| (h) | $NADP^+ + 2\,e^- \rightarrow NADP^-$ | 3.02 | 3.01 | 3.00 | 3.00 |

## Energetic comparison to the light-dependent reaction

The moderate barriers computed for the reaction mechanism raise an intriguing question: why does the light-dependent enzyme require an external driving force, as a quantum of light, to catalyze the reduction of the $C_{17}$=$C_{18}$ bond in PChlide by NADPH? We have therefore compared the energies and redox potentials of several reaction intermediates to those of other biochemical redox models (Table 6). Direct comparison of the redox potential of NADPH (Table 6, line f) to that of the two-proton, two-electron conversion of PChlide to Chlide (Table 6, line d) shows that the PChlide reduction to Chlide by NADPH should be thermodynamically favored in the ground state. Further analysis of the computed redox data shows that hydride transfer from NADPH (Table 6, line f) to PChlide to yield a *singly*-protonated, two-electron reduced PChlide (Table 6, line g) is thermodynamically disfavored, which entails that the spontaneity of the overall process arises from the addition of the second proton. Furthermore, the energetic barrier for the direct hydride transfer in the ground state has previously been shown to be very high ($>30$ kcal mol (*Heyes et al., 2009*; *Silva & Ramos, 2011*)), even after accounting for quantum tunneling effects (*Silva & Ramos, 2011*). Ground-state hydride transfer from NADPH to the substrate therefore may only occur if it precedes the protonation event. Indeed, hydride transfer from NADPH to $C_{18}$-protonated PChlide (Fig. 8) should proceed efficiently with a negligible barrier (1.0 kcal mol$^{-1}$) and very high exergonicity ($-33$ kcal mol$^{-1}$), but the actual feasibility of this step depends on the relative abundance of the $C_{18}$-protonated PChlide. Our computations on the dark-dependent PChOR, above, showed that the initial protonation of the $C_{17}$=$C_{18}$ bond by carboxylic acids (the most acidic amino acid side chains present in proteins) is thermodynamically expensive by 20 kcal mol$^{-1}$, which means that the natural abundance of $C_{18}$-protonated PChlide is very small ($e^{-20\,\text{kcal/mol/RT}}$). The overall barrier for the reduction of PChlide to Chlide by NADPH would therefore amount to at least those 20 kcal mol$^{-1}$ + the 1.0 kcal mol$^{-1}$ barrier for the hydride transfer when the initial PChlide protonation is performed by a carboxylic acid (like Asp or Glu) and would be even larger when weaker acids are used (like the Tyr or Lys residues actually present in the light-dependent PChOR active site (*Wilks & Timko, 1995*). Reduction of PChlide by NADPH therefore has too large an activation barrier to proceed at reasonable rates in the electronic ground state. The experimentally-observed initiation of the reaction upon uptake of a 590 nm photon (*Griffiths, McHugh & Blankenship, 1996*) can be easily computed to correspond to an increase of 1.05 V in the reduction potential of PChlide, which places it above the redox potential of NADPH and therefore makes the electron-transfer thermodynamically favorable.

More exotic pathways for ground-state reduction of PChlide by NADPH may also be excluded from consideration: for example, a two-electron transfer from NADPH to PChlide followed by energetically favorable $H^+$ transfer is also unfeasible, as the E°of the NADPH/NADPH$^{2+}$ pair (Table 6, line a) lies even farther above the PChlide substrate. On the other hand, two-electron transfer from a hypothetical (deprotonated) NADP$^-$ species (Table 6, line h), to PChlide (Table 6, line d), might be very favorable but is ruled out by the extreme difficulty of deprotonating NADPH: indeed, the difference of energies between

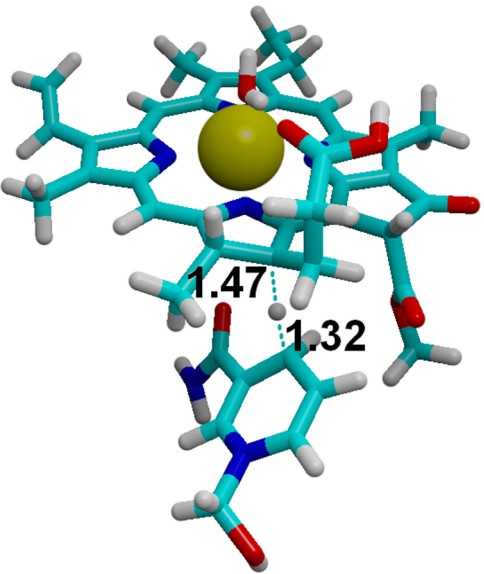

**Figure 8 Hydride transfer from NADPH to $C_{18}$-protonated PChlide.** The transition state for the hydride transfer from NADPH to $C_{18}$-protonated PChlide. Highlighted distances in ångstrom.

NADPH and $NADP^{-}$ in water (337.2 kcal $mol^{-1}$) is far higher than computed for even moderately weak acids (e.g., the difference between phenol and phenoxide (*Silva, 2009*) is only 299.2 kcal $mol^{-1}$), which implies an extremely high pKa for the NADPH proton.

The preceding analysis explains the need for an external energetic event for the reduction of protochlorophyllide by NADPH. Incidentally, our comparative analysis also showed that the two-electron/two-proton reduction of the double bond in PChlide, (Table 6, line d) is approximately as favorable as the comparable reduction of the typical C–C double bond found in fumaric acid (Table 6, line c), whereas the absence of $Mg^{2+}$ ion from PChlide disfavors this reduction process (Table 6, line e), which may explain why $Mg^{2+}$ becomes inserted into the porphyrin ring before the reduction of the $C_{17}$=$C_{18}$ bond.

## CONCLUSIONS

We have analyzed the proton and electron transfer events in light-independent protochlorophyllide oxidoreductase using medium-sized models. The reaction mechanism begins with single-electron reduction of the substrate by the $(Cys)_3$Asp-ligated [4Fe–4S] yielding a negatively-charged intermediate which, depending on the rate of Fe–S cluster re-reduction, either receives two protons before the final reduction event or receives a proton from the propionic side chain present on ring D, is reduced by a second electron and then abstracts a proton from Asp274 in a barrier-less process. The energetic barrier of the second alternative lies well below the experimental values, which suggests that the rate-limiting step *in vivo* is most likely to reside in the ATP-dependent re-reduction (*Kondo et al., 2011*) of the $(Cys)_3$Asp-ligated [4Fe–4S] by the L-protein, or in the reduction of the L-protein by cytoplasmic electron donors, which we have not attempted to address. Additional consideration of the protein environment allowed the confirmation of the broad features of the reaction mechanism, revealed a hitherto unsuspected

pH-dependence of the reaction potential of the Fe–S cluster and afforded valuable insights on the influence of specific amino acids on each reaction step.

The proposed reaction mechanism is made possible due to the low redox potential of the electron-donating (Cys)$_3$Asp-ligated 4Fe–4S cluster. In the light-dependent PChOR, this low-potential cluster is absent, and NADPH (which has a higher redox potential) is used as the electron donor. All possibilities of electron/hydride transfer from NADPH to PChlide were shown by our computations to be highly disfavored, clearly showing the reason behind the requirement for a quantum of light in the NADPH-dependent protochlorophyllide oxidoreductase, as it provides the energy needed to overcome this thermodynamically disfavored process by generating a more easily reducible state (*Silva & Ramos, 2011*) of the PChlide substrate.

### Funding

Research at REQUIMTE is supported by Fundação para a Ciência e a Tecnologia through grant no. PEst-C/EQB/LA0006/2011. This work has been financed by FEDER through Programa Operacional Factores de Competitividade–COMPETE and by Portuguese Funds through FCT–Fundação para a Ciência e a Tecnologia under project PTDC/QUI-QUI/111288/2009. The funders had no role in study design, data collection and analysis, decision to publish, or preparation of the manuscript.

### Grant Disclosures

The following grant information was disclosed by the author:
Fundação para a Ciência e a Tecnologia: PEst-C/EQB/LA0006/2011.
FEDER.
Portuguese Funds: PTDC/QUI-QUI/111288/2009.

### Competing Interests

The authors declare there are no competing interests.

### Author Contributions

- Pedro J. Silva conceived and designed the experiments, performed the experiments, analyzed the data, contributed reagents/materials/analysis tools, wrote the paper, prepared figures and/or tables.

### Data Deposition

The following information was supplied regarding the deposition of related data:
Output files of the Monte Carlo simulations have been deposited in FigShare:
Silva, Pedro (2014): Dark-operating PChOR. figshare.
http://dx.doi.org/10.6084/m9.figshare.1063691.
Retrieved 19:12, Aug. 02, 2014 (GMT).

**PeerJ** _______________________________________

## Supplemental Information

Supplemental information for this article can be found online at http://dx.doi.org/10.7717/peerj.551#supplemental-information.

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
