# Peer review of "With or without light: comparing the reaction mechanism of dark-operative protochlorophyllide oxidoreductase with the energetic requirements of the light-dependent protochlorophyllide oxidoreductase"

_PeerJ, doi:10.7717/peerj.551_

## Round 0.1 · original submission · Major Revisions

Please provide responses to the comments. Additional experiments may be needed.

·

Basic reporting

In general the manuscript fulfills all basic reporting standards of PeerJ, only two minor details should be addressed by the author:

1- Page 6: It would be appropiate to cite a reference for the 0.9804 scaling factor for vibrational frequencies used.

2- Table 1, 2 and 3, and Figure 3: In the indication of the level of theory used, it would be better to explicitly indiate the use of Grimme’s D3 dispersion correction.

Experimental design

No Comments

Validity of the findings

No Comments

Additional comments

The manuscript by Silva presents a detailed analysis of the H+ and e- transfer events in the light-indepedent protochlorophyllide oxidoreductase enzyme. In particular, the author addresses the reaction mechanism for the two H+ and e- transfer events leading to the product by carefully considering all possible reaction mechanisms, where the H+ and e- transfers occur as alternative succesions of events. The author uses sound methodologial methods and the study provides strong evidence for a proposed mechanism, where the suggested rate limiting step is the re-reduction of the 4Fe-4S cluster. Overall the study is complete, well-written and clear, and provides interesting insights on the reaction mechanism of this enzyme. I have a minor concern that the authors should consider. Once this point is addressed, I reccommend its publication in PeerJ.

1- The author explains that second-shell aminoacids affect proton-transfer energies by less than 2 kcal/mol, thus justifying the choice of model system used in the investigation of the reaction mechanism. However, long-range protein electrostatic effects from amino acids lying further could also potentially impact the computed energies. Moreover, a part from their impact on proton-transfer energies, such long-range effects due to specific charged protein residues could also have nonngeligible effects on e- transfer energies and redox potentials. Has the author also checked the effect of second-shell amino acids on the computed e- transfer energies? In addition, could the author justify the validity of the continuum solvent assumption used to describe protein solvation effects, in contrast e.g. to QM/MM approaches, may be refering to other studies addresing this issue?

·

Basic reporting

The paper presents an interesting and detailed computational study on the energetic requirements of the enzymatic protochlorophyllide (Pchlide) two-electron reduction. Two reactions, both occurring in living organisms, one involving an iron-sulfur cluster and the other involving NADH as reducing reagents are analysed and compared. The computational analysis of the iron-sulfur reduction reaction is done for the first time, and is highly interesting for a wide research community (which broad research topics such as enzymatic redox reactions, chlorophyll biosynthesis and regulation, iron-sulfur-cluster catalysis and computational-chemistry for biochemical applications).

Experimental design

The analysis is based on the evaluation of the redox potentials and electron- and proton transfer activation energies with the help of the density-functional theory calculations. The energy barriers of all relevant proton-transfer reactions in various redox states are also determined. The computational data allows for discussing various pathways; the author argues that re-reduction of the iron-sulfur cluster is likely to be the rate-limiting step in the previously reported in vitro studies. In such a case, the Pchlide reduction under dark conditions is initiated by electron transfer from the iron-sulfur cluster followed by a protonation step from the protein (Asp274) rather than from the substrate itself (the propionate). A similar analysis performed for the NADH reduction indicates that the reaction is likely to have a rather high energy barrier, which is consistent with the conclusions of the previous computational studies and which also validated the approach taken in the current study.

Validity of the findings

Considering the wide community interested in this study, I suggest to the author to conduct a revision of the manuscript in order to provide some missing details and descriptions to facilitate reading and understanding of the paper. I list my comments to assist the revision in the "General Comments to the Author" subsection.

Additional comments

"Computational methods":

1. Please describe how the saddle points were located.

2. please specify the charge and multiplicity of the iron-sulfur cluster in the two redox states in the calculations. I would also suggest presenting its structure in a figure.

3. It is unfortunate that In Scheme 1, the computed energies (the reaction energy and the reorganization energies) are not indicated. it is unclear which energies (I assume that the two reorganization energies??) were used to define the Marcus parabolas? These details should be clarified for the reader also to facilitate understanding of the results presented on page 16.

4. The formula sequence on pages 7 and 8 is difficult to follow. In addition to what is given, it would be helpful to present the resulting formula in which the computed DFT energies were plugged. Also specify how the energies were computed or make sure that it is mentioned in the results which energies of Tables 1 and 2 were used to derive certain potentials and electron transfer barriers.

"Results and discussion"

5. the content of Table 1 is difficult to understand. For instance, it should be explained that "X...Y" corresponds to the energy of a proton-transfer transition state. If I understood it correctly, the energies from Table 1 were used to produce the energy diagram in Figure 3. The Figure is much easier to follow; my suggestion is to modify the description to refer first to the Figure, and then to give the reference to the table as containing supplementary information to the Figure. Thus, I would recommend change the ordering in the text to Figure 3, Table 1 and Figure 2.

6. The description of the electron-transfer energy-barrier determination should be extended. In the current version, I could not follow these results but they are of central significance. The supplementary information presenting the Marcus curves was also not very helpful. I suggest, that an explicit description of one of such step barriers (e.g., the initial electron-transfer step, which is the most relevant for the discussed mechanism) is included in the results and discussion part together with a figure showing the derived Marcus parabolas.

7. Table 2: what do "n.d." and "n.a." mean?

8. In the discussion on page 21, could the author specify which redox potential and by how much changes upon photoexcitation of the Pchlide? Adding this point to the discussion would be in line with the chosen methodology.

Minor corrections:

page 4, line 17: "density-function theory methods";

page 5, line 3: avoid saying "uninteresting", a more specific sentence concerning the role of the hydrophobic side-chains (if at all) would be more appropriate;

page 9, line 2: I would omit "of its reaction mechanisms"; its very clear what a "rate determining step" referrers to. Here it is more appropriate to talk about a complex chemical reaction than about an enzymatic-reaction mechanism, which also includes for instance substrate binding and release.

page 12, line 1: not "electronic energies" but "total energies", unless the vertical energy estimates are discussed. May be further clarification is required here.

page 15 lines 1-3: too many "easy" words;

page 15 lines 16-18 the phrase in brackets is too difficult to understand; (may be better to refer to the particular step of the reaction rather than to identify the reactant and product; referring to the specific data in the table may also help).

page 16, line 4: what about "redox state" instead of "extra electrons present"?

page 16 lines 4-6: the sentence is easy to misinterpret: Strictly speaking, the redox state of the iron-sulfur cluster matters for the protonation reaction as Figure 3 clearly demonstrates. I would omit this sentence.

page 16 line 25: omit "very".

Reviewer 3 ·

Basic reporting

In this paper the author investigates the reaction mechanism of the dark-operative protochlorophyllide oxidoreductase using two small DFT models. One contains the protochlorophyllide cofactor and the side chain of Asp274, while the other only describes the Fe4S4 cofactor.
This is the main problem of the article. The models are too small to correctly account for the enzymatic mechanism.
It is unclear how the pKa of the aspartate is shifted for this residue to be protonated or what is the influence of the protein on the redox potential of the FeS cluster.
A new and larger model should be envisioned that correctly describes the enzyme.

Experimental design

No comments

Validity of the findings

No comments

---

## Round 0.2 · accepted · Accept

Thank you for your submission to PeerJ. I am writing to inform you that your manuscript, "With or without light: comparing the reaction mechanism of dark-operative protochlorophyllide oxidoreductase with the energetic requirements of the light-dependent protochlorophyllide oxidoreductase" (#2013:02:312:1:1:REVIEW), has been accepted for publication.